# Molecular Characterization of a Restriction Endonuclease PsaI from *Pseudomonas anguilliseptica* KM9 and Sequence Analysis of the PsaI R-M System

**DOI:** 10.3390/ijms26146548

**Published:** 2025-07-08

**Authors:** Beata Furmanek-Blaszk, Iwona Mruk, Marian Sektas

**Affiliations:** Department of Microbiology, Faculty of Biology, University of Gdansk, Wita Stwosza 59, 80-308 Gdansk, Poland; iwona.mruk@ug.edu.pl (I.M.); marian.sektas@ug.edu.pl (M.S.)

**Keywords:** restriction–modification systems, type II restriction endonuclease PsaI, isoschizomer HindIII

## Abstract

A restriction enzyme PsaI, an isoschizomer of the type II restriction endonuclease HindIII, has been purified to homogeneity from Gram-negative bacilli *Pseudomonas anguilliseptica* KM9 found in a wastewater treatment plant in Poland. Experimental data revealed that R.PsaI is highly active in the presence of Co^2+^, Mg^2+^, and Zn^2+^ and reached a maximal level of activity between 2.5 and 10 mM while its activity was significantly decreased in the presence of Ca^2+^, Fe^2+^, Mn^2+^, and Ni^2+^. Moreover, we found that the purified R.PsaI did not require NaCl for enzyme activity. Restriction cleavage analysis followed by sequencing confirmed 5′-AAGCTT-3′ as the recognition site. The genes for restriction–modification system PsaI were identified and characterized. Downstream of the psaIM gene, we noticed an ORF that shares extensive similarity with recombinase family protein specifically involved in genome rearrangements. Sequence analysis revealed that the PsaI R-M gene complex showed striking nucleotide sequence similarity (>98%) with the genes of the PanI R-M system from a *P. anguilliseptica* MatS1 strain identified in a soil sample from Sri Lanka.

## 1. Introduction

The two enzymes that make up type II restriction–modification (R-M) systems are a methyltransferase, which transfers a methyl group to a specific nucleotide to prevent cleavage, and a restriction endonuclease, which identifies and cleaves a particular DNA sequence at a specific site. Four groups of restriction endonucleases have been identified based on the DNA recognition sequence, cleavage position, cofactor requirements, and protein subunit composition [1]. Type II enzymes have been studied extensively because they play a fundamental role in gene analysis and cloning work. Most of them require divalent metal ions to catalyze the cleavage of the phosphodiester bond. Although the natural cofactor for all type II restriction endonucleases is Mg^2+^, they can utilize a variety of divalent cations for in vitro DNA cleavage reaction, including Ca^2+^, Cd^2+^, Co^2+^, Fe^2+^, Mn^2+^, Ni^2+^, or Zn^2+^ [2,3].

Many R-M enzymes have been isolated from various microorganisms over the years, but the necessity of restriction endonucleases with unique specificities for molecular cloning has largely motivated the search. Because type II restriction enzymes can identify and cut DNA sequences and produce a predictable cleavage pattern, they are most frequently utilized in molecular biology applications. This makes them essential for recombinant DNA technology [4].

Among the over 4000 biochemically or genetically characterized restriction enzymes found to date, 98% belong to type II [4]. This high percentage could result from their great demand and ease of detection, although it might not accurately represent the distribution of R-M systems. Type II restriction endonucleases generally recognize the palindromic sequence in DNA and cleave within or near the recognition sequence, producing DNA fragments of defined sizes. These properties have led to the screening of diverse bacterial species for new restriction activities. Up to now, numerous target sites have been recognized by various restriction endonucleases, many of which have new specificities that have not yet been found.

In this work, we investigated samples taken from water pollution control plants at each stage of sewage treatment. Bacterial communities found in wastewater treatment plants (WWTPs) can be quite varied, consisting of hundreds of species [5]. In this heterogeneous ecosystem, bacteria-dominant microorganisms belong to the phyla Firmicutes, Proteobacteria, and Bacteroidetes [6]. *Pseudomonas* spp. are able to thrive in diverse ecological niches, including highly nutritious environments, such as sewage [7].

High abundance, density, heterogeneity, activity, and complex interplay within activated sludge bioreactors, like those in WWTPs, would appear to be linked to increased rates of gene flow, including both horizontal and vertical transfer of genes. Taking into account the vast amount of bacteria passing through WWTPs every day, it is likely that many of them could serve as a potential source of various enzymes. Sludge is also increasingly being viewed and treated as a source of valuable new products originating from bacteria.

To date, many restriction endonucleases have been found in various bacteria, some of which are isoschizomers used as tools in molecular biology. In this work, we investigated samples from the wastewater treatment plant “Gdynia-Debogorze”, and the isolated bacteria were screened for the presence of restriction endonucleases. One of the isolates found in the activated sludge, identified as *P. anguilliseptica* KM9, produced the restriction endonuclease PsaI. In this manuscript, we describe the isolation, purification, and characterization of the enzyme and analyze its properties. R.PsaI belongs to type II restriction endonucleases, and it is an isoschizomer of the HindIII enzyme from *Haemophiluus influenzae*. The *P. anguilliseptica* KM9 strain seems to be a promising source of R.PsaI possessing HindIII-type restriction activity.

## 2. Results and Discussion

### 2.1. Detection of Restriction Endonucleases Activity in Bacterial Isolates

WWTPs are the habitat of many types of microorganisms, which makes them interesting considering their potential to produce important enzymes for biotechnological exploration. Furthermore, the production of restriction endonucleases in this particular environment provides bacterial cells with powerful protection against a broad spectrum of phages. Years of screening various microbial sources have yielded hundreds of restriction enzymes that recognize and cleave specific DNA sequences. Knowing that R-M systems have been detected in bacteria from all ecological niches and taxonomic groups, including WWTPs, we aimed to screen this microbiome resource and search for novel microbial restriction endonucleases. In this work, a good method to screen for type II restriction endonucleases consisted of incubation on cell extracts with known DNA substrates [8].

By employing the quick screening method to analyze the restriction activities in the crude cell extracts, restriction endonucleases were found in at least 18 of the 320 isolates. Only one strain, identified as *P. anguilliseptica* KM9, showed restriction activity, which gave a clear, sharp, banding pattern on lambda DNA similar to that observed with lambda genome digested with HindIII restriction endonuclease. Hence, it was selected for further study (Appendix A). According to the suggested nomenclature rules, this enzyme was named PsaI [9].

### 2.2. Enzyme Purification

Because our laboratory has been involved in studies to characterize R-M systems, we report here the isolation and characterization of R.PsaI. The purification process involved four chromatographic steps. The crude cell extract contained a set of contaminating proteins, and most of these were eliminated through phosphocellulose chromatography. The strategy used for purifying R.PsaI is summarized in Table 1.

The final enzyme preparation was free of non-specific nucleases. From 10 g of bacteria, we were able to obtain 1.35 mg of homogenous enzyme preparation, with an overall yield of 4.5%. R.PsaI was analyzed through SDS-PAGE to determine its purity and denaturated molecular mass. According to Coomassie blue-stained gels, the enzyme was found to be at least 95% pure. Relative to the standards of a known M*r*, a value of M*r* = 35,000 ± 100 for R.PsaI was calculated (Appendix A). Enzyme stability was tested after different periods of storage at −20 °C; the purified enzyme did not lose activity after storage for 12 months in buffer with 50% glycerol.

The simplicity of cultivation, presence of the single restriction endonuclease, and high level of its production make *P. anguilliseptica* KM9 a promising producer of PsaI restriction endonuclease isoschizomeric to R.HindIII for use in experimental practice in industry. To date (14 May 2025), 269 isoschizomers of HindIII restriction endonucleases have been found in many bacterial genera. However, only one has been found in the genus *Pseudomonas* [4]. They are reported in the Restriction Enzyme database (http://rebase.neb.com) (accessed on 14 May 2025), although only R.HindIII is commercially available from fourteen suppliers.

### 2.3. Optimal Conditions for Restriction Activity

The effect of several factors on R.PsaI activity was investigated. To identify the optimal buffer and temperature, the enzyme was tested for its ability to digest 150 ng of lambda DNA substrate with one unit of the enzyme in one hour. The purified enzyme was active over a temperature range of 30–52 °C (Figure 1).

When the cleavage activity of R.PsaI was tested in different commercial buffers using lambda DNA as a substrate, the formation of a characteristic cleavage pattern was observed in all of the tested buffers. Upon digestion of lambda DNA with R.PsaI for 24 h, no traces of unspecific cleavage products were noted.

### 2.4. Effect of Divalent Metal Ions and Ionic Strength on R.PsaI Activity

The activity of most enzymes is easily affected by environmental factors, such as metal ions that are usually found in wastewater. Therefore, one of the objectives of this study included investigating the effect of various concentrations of some metals, referred to as trace elements, on the activity of the restriction enzyme we isolated from *P. anguilliseptica* KM9. Type II restriction endonucleases require certain cofactors to digest DNA. Almost all restriction endonucleases of the PD-(D/E)XK family need divalent cations—usually Mg^2+^, which is a natural cofactor for the majority of these enzymes—for DNA cleavage [10]. Although the physiological metal ion for the restriction endonucleases is magnesium, they can utilize a variety of divalent cations for in vitro cleavage reaction [11]. Firstly, to identify whether other metal ions can promote endonucleolytic activity of R.PsaI, we performed an array of reactions using different divalent cations. To determine R.PsaI activity in the presence of various divalent cations, such as Ca^2+^, Co^2+^, Fe^2+^, Mg^2+^, Mn^2+^, Ni^2+^, or Zn^2+^, we used plasmid pUC18 DNA containing a single recognition site as a substrate and then analyzed the products through gel electrophoresis. The results showed an absolute requirement for divalent cations for R.PsaI cleavage activity, preferably Co^2+^ and Mg^2+^ (Figure 2). Four metal ions (calcium, iron, manganese, and nickel) failed to support cleavage, whereas zinc displayed only a small reduction in cleavage activity.

Many type II restriction endonucleases can use Mn^2+^ in place of Mg^2+^ as a cofactor, but only a few can use a broader range of divalent cations instead [3].

To investigate more precisely the requirements of R.PsaI, we carried out DNA cleavage experiments at various divalent cation concentrations ranging from 2.5 to 10 mM in the presence of plasmid pUC18 DNA containing a single recognition site as a substrate, and then we analyzed the products through gel electrophoresis (Appendix A). R.PsaI shows a qualitatively similar Co^2+^, Mg^2+^, and Zn^2+^ concentration dependence. In general, the maximum level of activity was observed in the presence of Mg^2+^, Co^2+^, and Zn^2+^ at concentrations between 5 and 10 mM. We also noticed that R.PsaI showed a slight preference for higher Co^2+^ concentrations compared to Mg^2+^ or Zn^2+^ (between 2.5 and 10 mM) to maintain the same cleavage efficiency.

Our results suggest that R.PsaI has a metal-binding site that can accept varied divalent metal ions, modulating the catalytic activity. In industrial wastewaters, various metal ions are present, but only some of them are relevant in the environmental context. The ability to utilize a wide range of metal cofactors for DNA cleavage may be of great importance to the biological function of R.PsaI residing in *P. anguilliseptica* KM9. When fighting with other bacteria for their environment and phages that have the potential to infect the host cell, this characteristic may help the bacteria break down invasive foreign genomes.

Restriction enzymes show different specificity in their response to ionic strength. The effect of ionic strength on R.PsaI activity was determined by varying the NaCl concentrations from 0 to 250 mM. The enzyme is remarkably tolerant to high concentrations of NaCl, being active up to 200 mM NaCl. The activity decreased at a concentration higher than 200 mM (partial DNA digestion) (Figure 3). Moreover, in the absence of NaCl, lambda DNA cleavage was comparable to that observed in the presence of a wide range of sodium chloride concentrations.

R.PsaI retains activity with or without NaCl and thus differs from other restriction endonucleases, which require salt to remain active. For that reason, R.PsaI could be a valuable tool for DNA manipulation as it is active at a broad range of salt concentrations.

### 2.5. Determination of the R.PsaI Cleavage Site

To determine the recognition sequence and cleavage site of R.PsaI, plasmid pGEM3Zf(+) was digested with purified R.PsaI. Then, the linear form of DNA was sequenced using GS5 and GS6 primers (Appendix A) (Figure 4). The sequencing of linear plasmid DNA revealed the recognition sequence 5′-AAGCTT-3′, where the cleavage occurs between two adenines.

### 2.6. Genomic DNA Modification Status

The results described above prompted us to examine the pattern of DNA methylation. The presence of cognate methyltransferase activity was assessed based on its ability to protect genomic DNA from cleavage by the PsaI restriction enzyme. *P. anguilliseptica* KM9 displayed methylation protection against its endogenous restriction enzyme (Figure 5), as well as isoschizomers of R.HindIII, while digestion by restriction endonucleases with different specificities resulted in DNA cleavage and subsequent degradation (Figure 5).

The data revealed the existence of bacterial methyltransferase activity, which suggested the presence of a complete type II restriction–modification system in *P. anguilliseptica* KM9 cells. It should be noted that this protection against R.PsaI only indicates the possibility, but does not prove, that M.PsaI has identical specificity.

This finding prompted us to screen the *P. anguilliseptica* KM9 DNA sequence for the presence of a gene coding for cognate PsaI methyltransferase. Studies with R-M systems revealed that restriction endonucleases show little primary sequence similarity among themselves, whereas methyltransferases share substantial sequence homology and could be identified based on primary sequence data [12]. Knowing that the greatest similarity between methyltransferases of R-M systems is restricted to the two most conserved motifs (I and IV) [13,14], we designed primers for amplification of a potential methyltransferase PsaI gene containing conserved motifs I and IV. The forward primer (isometDIPY) (Appendix A) was complementary to the S-adenosylmethionine binding motif (I), whereas the reverse primer (isometR) (Appendix A) was complementary to the methylation catalytic motif (IV). To determine the partial nucleotide sequence of the gene coding for M.PsaI, genomic DNA of *P. anguilliseptica* KM9 was subjected to PCR amplification of the selected methyltransferase coding DNA fragment. This pair of primers produced an amplicon at a size of around 700 bp, showing sequence similarity to site-specific DNA methyltransferases. The presence and distribution of highly conserved amino acid sequence motifs indicated that M.PsaI is a member of the N^6^-adenine methylases family. Because genes encoding R-M systems are usually closely linked, we decided to analyze DNA fragments located upstream and downstream of the identified *psaIM* gene fragment.

For this, the iPCR approach, which allows for amplification of the unknown sequences adjacent to the known DNA fragment, was employed. For this purpose, we designed primers isometF and isometinv1 (Appendix A) oriented in opposite directions. Nucleotide sequencing of the obtained PCR product revealed the presence of a DNA fragment at a size of around 800 pb showing sequence similarity to type II restriction endonuclease HindIII. Using this technique, we determined the nucleotide sequence of a 1.8-kilobase DNA stretch containing two incomplete open reading frames showing significant similarity to genes coding for restriction and modification enzymes isospecific to R-M system HindIII. However, for an unknown reason, using the iPCR technique, we were unable to amplify either the 5′ or 3′ termini of *psaIR* and *psaIM* genes.

In 2022, an article describing the isolation and partial characterization of a new restriction endonuclease PanI, isoschizomer of R.HindIII, from the *P. anguilliseptica* MatS1 strain was published [15]. We noticed that the *panIR* gene shared a striking similarity to the known portion of the *psaIR* gene we identified in the *P. anguilliseptica* KM9 strain. Knowing that genes that code for restriction enzymes nearly always occur next to the genes that code for the corresponding methyltransferases, we analyzed sequences surrounding the *panIR* gene and noticed that the methyltransferase gene *panIM* lies immediately adjacent to the *panIR* gene. This finding indicated the presence of a complete R-M system in the *P. anguilliseptica* MatS1 strain.

Then, in order to identify the lacking sequences corresponding to the 5′ and 3′ ends of *psaIR* and *psaIM* genes, respectively, we designed primers based on published sequences of genes encoding the R-M system PanI [15] (resPsastart and metPsaend) (Appendix A). Nucleotide sequencing of the obtained PCR products uncovered the presence in *P. anguilliseptica* KM9 of an R-M gene complex almost identical to R-M system PanI. Protein alignment results have shown that R.PsaI and M.PsaI share about 98% identity with their bacterial counterparts of *P. anguilliseptica* MatS1.

### 2.7. PsaI R-M System Analysis

In this work, we also defined R-M system PsaI residing in the *P. anguilliseptica* KM9 strain. The PsaI R-M system is organized in a head to tail orientation often found in many R-M systems. Among a group of R-M systems isospecific to HindIII, only the prototype R-M system from *Haemophilus influenzae* Rd has a similar genetic organization (Figure 6) [16].

The G + C content of the genes encoding the complete PsaI R-M system is 43.73%, which is lower than the overall guanine–cytosine composition of *P. anguilliseptica* genomes (60%) [17]. The presence of R-M genes with different GC content suggests that they could have been integrated into bacterial genomes through horizontal gene transfer. The restriction endonuclease and modification methylase genes lie adjacent to each other and are oriented transcriptionally in a sequential manner. The start codon of M.PsaI is placed just before the stop codon of R.PsaI. The first gene of the PsaI R-M system, *psaIR*, encodes a protein with a calculated molecular mass of 34,633 Da, which consisted of 304 amino acids. Database searches showed that this protein, apart from a remarkable similarity to R.PanI (98%) [15], also shares significant homology to the HindIII family type II restriction endonucleases found in many bacteria. The greatest level of similarity (≥70%) was observed between R.PsaI and proteins from *Marinospirillum* sp. (78.62%), *Cylindrospermopsis raciborskii* (75.66%), *Candidatus Nitrotoga fabula* (76.32%), and *Negativicutes bacterium* (73.36%) (for details, see [15]). The alignment of the R.PsaI amino acids sequence with well-studied isospecific restriction endonucleases of R-M systems, such as R.EcoVIII, R.LlaCI, R.Csp231I, and R.HindIII, revealed homology of 56.62%, 28.57%, 21.89%, and 18.26%, respectively [16,18,19,20] (Figure 7A). All of the above-mentioned restriction endonucleases are members of the PD-(D/E)XK superfamily of Mg^2+^-dependent nucleases (Figure 7A) [10].

The second gene, *psaIM*, encodes a polypeptide of 306 amino acid residues (M.PsaI) with a calculated mass of 34 517 Da containing conserved amino acid sequence motifs typical for the *N^6^*-adenine β-class methyltransferases [13]. The alignment of the entire M.PsaI amino acids sequence with well-studied isospecific methyltransferases of R-M systems, such as M.EcoVIII, M.LlaCI, M.Csp231I, and M.HindIII, revealed significant homology of 64.47%, 57.77%, 57.47%, and 49.34%, respectively [16,18,19,20]. (Figure 7B). In addition, M.PsaI also shares remarkable sequence identity with methyltransferases from *Marinospirillum* sp. (80.27%), *Candidatus Nitrotoga fabula* (79.25%), *Negativicutes bacterium* (78.86%), and *Cylindrospermopsis raciborskii* (77.85%).

Bioinformatic analysis revealed that a large number of R.HindIII homologs are distributed in several bacteria. However, only a few HindIII-type R-M systems have been well-characterized in *H*. *influenzae* (HindIII), *Citrobacter* sp. (Csp321I), *Lactococcus lactis* (LlaCI), *Bacillus stearothermophilus* (BstZ1II), and *Escherichia coli* (EcoVIII) [16,18,19,20].

### 2.8. Analysis of the Regions Flanking the R-M System PsaI

The flanking regions of R-M systems are often characterized by the occurrence of mobile genetic elements involved in recombinational events [21,22]. This is also the case for the PanI R-M system. Analysis of the nucleotide sequence revealed an ORF, located immediately 5′ of the *panIR* gene, that encodes the magnesium chelatase domain-containing protein, whereas downstream of the *panIM* gene an ORF that is related to a mobility-associated protein (recombinase family protein) is placed. Having determined a complete PsaI R-M system, we made an attempt to identify sequences adjacent to the *psaIM* and *psaIR* genes. For this purpose, two sets of primers were designed; in each pair, one primer was targeted at a known region of R-M system PsaI, while the second primer matched the sequence flanking isomeric R-M system PanI (Appendix A). Using a primer set complementary to *psaIM* and a recombinase gene adjacent to *panIM,* we obtained a PCR product of the expected size (900 bp) and determined its sequence. An examination of this DNA fragment revealed the presence of an ORF coding for recombinase that is almost identical (98%) to its counterpart in the PanI R-M system. However, when the second set of primers designed for *psaIR* and the magnesium chelatase domain-containing protein gene adjacent to *panIR* were involved, no band of the expected length was detected.

This observation suggests that a homologous DNA segment containing the R-M complex is present at different chromosomal loci in the two *Pseudomonas* genomes. We hypothesized, therefore, that the presence of a recombinase enzyme involved in DNA mobility in the vicinity of the R-M PsaI system may serve to transfer the R-M gene complex within and between bacteria. Indeed, there have been reports indicating that the presence of a gene encoding recombinase near genes of the R-M system can be an indicator of horizontal gene transfer [21,22]. Additionally, the migration of DNA segments containing R-M complexes at different positions in the genome may be facilitated by extensive genomic plasticity and diversity of the genus *Pseudomonas* [7].

It should be noted that both *P. anguilliseptica* strains have almost the same nucleotide sequence within their R-M systems. Moreover, they are geographically unrelated and have been recovered from different environments. Strain KM9 was isolated from the activated sludge of a WWTP in Poland, while the MatS1 strain was identified in a soil sample from Sri Lanka.

## 3. Materials and Methods

### 3.1. Site Description and Sampling Protocol

The WWTP in Gdynia Debogorze receives and treats municipal sewage from Gdynia and other small surrounding towns. It serves about 470,000 people and discharges approximately 60,000 m^3^ per day to the sea. Activated sludge samples were collected in polypropylene tubes, transported to the laboratory at a temperature of 4 °C, and processed on the day of collection. Samples were collected in April, May, and October 2012.

### 3.2. Growth Conditions

Samples containing bacteria were inoculated onto MacConkey agar and incubated at 37 °C for 24 h under aerobic conditions. The next day, the specimens were examined for growth and colony morphology and subjected to Gram staining. One isolate producing a restriction enzyme that cleaves DNA yielding a characteristic pattern was selected for identification. A single colony of bacteria was identified as *P. anguilliseptica* using MALDI-TOF MS (MALDI biotyper; Bruker Daltonics, Billerica, MA, USA) according to the manufacturer’s instructions. This strain was routinely cultured at an optimum growth temperature of 25–27 °C in tryptic soy agar (TSA). The bacteria was maintained on TSA agar plates at room temperature and for long-term storage kept frozen at −70 °C in LB broth supplemented with 20% glycerol. For R.PsaI purification, a single colony of *P. anguilliseptica* KM9 was grown in LB broth at 30 °C for 24 h, and this culture was used for further inoculation of 1 L of LB. Erlenmeyer flasks were incubated at 30 °C for the next 24 h, and cells were harvested through centrifugation and stored frozen.

### 3.3. Restriction Endonuclease Activity Assay in Cell Lysate

The occurrence of restriction endonuclease in bacterial strains was tested using the modified lysozyme and Triton X-100 method [8]. Bacterial cells were collected from Petri dishes and transferred into 20 µL of incubation mixture A, containing 20 mM Tris-HCl pH 8.0, 1000 mM NaCl, 12.5 mM EDTA, 10 mM 2-mercaptoethanol (ME), and lysozyme at a concentration of 10 g/L. The sample was incubated for 30 min at room temperature, and then 20 µL of incubation mixture B containing 20 mM Tris-HCl pH 8.0, 2% Triton X-100, and 10 mM ME was added for 60 min at 6 °C. The restriction endonuclease activity was assayed in 20 µL of reaction mixture containing 0.1 µg of λ or pUC18 DNA, 2 µL of the restriction buffer Tango (Fermentas, Lafayette, CO, USA), and 2 µL of bacterial lysate cleared through centrifugation for 1 min (10,000× *g*). The universal Tango buffer was used because it generally yields high activity for most of the restriction enzyme. After incubation for 20 min at 37 °C, the mixture was treated with 20 µL of phenol to stop the reaction and extract the cleavage products, which were subjected to agarose gel electrophoresis. The fragmentation patterns obtained for DNA substrates cleaved with the enzyme present in the cell lysate were analyzed using the REBASE Tools REBsites [4], NEBcutter [23], and REBpredictor [24].

### 3.4. Purification of the Native Restriction Endonuclease PsaI

All of the purification steps were carried out at 4 °C, unless otherwise stated. The cell paste (10 g) was suspended in 30 mL of buffer P (10 mM potassium phosphate pH 7.0, 20 mM KCl,1 mM EDTA, 10 mM 2-mercaptoethanol (ME), 5% *v*/*v* glycerol) supplemented with 0.1 mM PMSF as a protease inhibitor and disrupted through sonication at 4 °C in 60 × 10 s bursts. The lysate was clarified through centrifugation (14,000 rpm, 30 min) and applied to a 2.5 × 7 cm phosphocellulose P11 column (Whatman, Little Chalfont, Buckinghamshire, UK) equilibrated with buffer P. The column was then washed with 300 mL of buffer P, and the proteins were eluted with a 200 mL KCl gradient (0.02–1.0 M) in the same buffer. Fractions of 3 mL were collected and assayed for PsaI endonuclease activity. The enzyme was eluted between 0.25 and 0.4 M KCl. The active fractions were then dialyzed against buffer H (10 mM potassium phosphate pH 7.0, 200 mM KCl, 10 mM ME, 5% *v*/*v* glycerol) loaded onto a 2.5 × 3 cm hydroxylapatite column (BioRad, Hercules, CA, USA) and eluted with 200 mL of linear gradient of K-phosphate, pH 7.0 (0.01–0.4 M). After that, the fractions with the highest endonucleolytic activity were dialyzed against buffer C (10 mM potassium phosphate pH 7.0, 20 mM KCl, 1 mM EDTA, 10 mM ME, 5% *v*/*v* glycerol), loaded onto a 2.5 × 22 cm CM Sephadex C-50 column (Pharmacia, New York, NY, USA), and the proteins were eluted with 250 mL of KCl gradient (0.05–0.8 M). The final preparations of the enzyme were dialyzed against a storage buffer containing 10 mM potassium phosphate, pH 7.8, 50 mM KCl, 0.1 mM EDTA, 10 mM ME, and 60% *v*/*v* glycerol and stored at −20 °C.

### 3.5. Molecular Mass Determination

Sodium dodecyl sulfate polyacrylamide gel electrophoresis (SDS-PAGE) was carried out in gels containing 10% acrylamide for the separation gel and 4% acrylamide for the stacking gel. After electrophoresis, the protein bands were visualized through Coomassie brilliant blue R 250 staining. The M*r* of the PsaI was calculated using a calibration curve obtained with the following standard proteins: bovine serum albumin (67 kDa), ovalbumin (43 kDa), carbonic anhydrase (30 kDa), and trypsin inhibitor (20.1 kDa).

### 3.6. DNA Cleavage Activity Assays

PsaI endonuclease activity was assayed in a 20 µL reaction mixture containing 2 µL of the restriction buffer Tango (Thermo Scientific, New York, NY, USA), 150 ng of λ phage DNA, and 2 µL of column fraction (1 h, 37 °C). The DNA was analyzed on 1% agarose gels in 1× TBE buffer, and the completeness of lambda DNA cleavage was evaluated visually. One unit of endonucleolytic activity was defined as the minimal amount of R.PsaI that completely digests 1 µg of phage λ DNA in 1 h at 37 °C.

### 3.7. Determination of the R.PsaI Recognition Sequence

Through restriction mapping of lambda and pUC18 plasmid DNAs, we were able to determine the recognition sequence of the purified endonuclease.

DNAs were digested into definite fragments using R.PsaI, and a double digestion reaction was also carried out using a second restriction enzyme, which enabled the localization of R.PsaI cleavage sites. The size of the DNA fragments generated through digestion was predicted using the REBpredictor program available on the website (http://tools.neb.com/REBpredictor/index.php) (accessed on 16 June 2014). The positions of putative recognition sequences were matched with the sites mapped through double endonuclease digestion. Then, the expected cleavage fragment was compared with the observed restriction fragments from R.PsaI cleavage of the DNAs.

To confirm the predicted recognition sequence and the cleavage site, R.PsaI-digested pGEM3Zf(+) plasmid DNA was used as a template for DNA run-off sequencing. Purified DNA was subjected to run-off Sanger automated sequencing (Genomed, Warsaw, Poland) using GS5 and GS6 primers (Appendix A). Sequencing data were analyzed using Chromas Lite version 2.6.6 software (Technelysium Pty Ltd., South Brisbane, Australia).

### 3.8. Characterization of the R.PsaI Activity Optima

The temperature dependence of R.PsaI activity was tested by incubating 150 ng of lambda DNA with one unit of the enzyme at different temperatures (30 °C–52 °C) for one hour. R.PsaI activity in buffers supplied with commercial restriction endonucleases from New England Biolabs (NEBuffer 1, 2, 3, and 4) and Thermo Scientific (TANGO, B, G, O, and R) was assessed by digesting 150 ng of lambda DNA with one unit of the purified enzyme for one hour at 37 °C. NaCl’s effect was studied using the same protocol in the presence of different NaCl concentrations (0–250 mM). The effects of divalent cations, such as Ca^2+^, Co^2+^, Fe^2+^, Mg^2+^, Mn^2+^, Ni^2+^, and Zn^2+^, on R.PsaI activity were investigated. Digestions were carried out in 20 µL reaction mixtures containing a buffer free of magnesium ions (50 mM NaCl, 10 mM Tris-HCl, 1 mM DTT, pH 7.9), 0.33 µg of pUC18 plasmid DNA, 10 mM of specific divalent cations, and a 0.5 unit of the enzyme (1 h, 37 °C).

### 3.9. Determination of the DNA Modification Status of the P. anguilliseptica KM9 Strain

To test the endogenous methyltransferase activity, genomic DNA of *P. anguilliseptica* KM9 was digested with PsaI restriction endonuclease and its isoschizomers, R.HindIII and R.EcoVIII. The reaction mixture had a final volume of 20 µL that contained 1 µg of lambda DNA and 1× appropriate NEB buffer, and it was incubated at 37 °C for one hour. The digestion products were subjected to 0.8% agarose gel electrophoresis.

### 3.10. Determination of the Complete Nucleotide Sequence of the PsaI R-M System

The procedure for inverse PCR (iPCR) was performed according to Ochman et al. [25]. The iPCR template was prepared through self-ligation of the Sau3AI-digested DNA from *P. anguilliseptica* KM 9. The resulting ligation products were amplified with primer pairs and sequenced. The sequences of the PCR products were assembled to reconstruct the sequence of a region of interest and submitted to the GenBank database (accession number PQ730136). The details of the primers used are shown in Appendix A.

### 3.11. Bioinformatic Tools

DNA and protein sequence similarity searches were performed using Nucleotide BLAST and Protein BLAST respectively [26], BLAST against REBASE [4], or CLUSTALW [27]. Queries for all available R-M systems were obtained from the Restriction Enzyme database (REBASE) website [4].

## Figures and Tables

**Figure 1 ijms-26-06548-f001:**
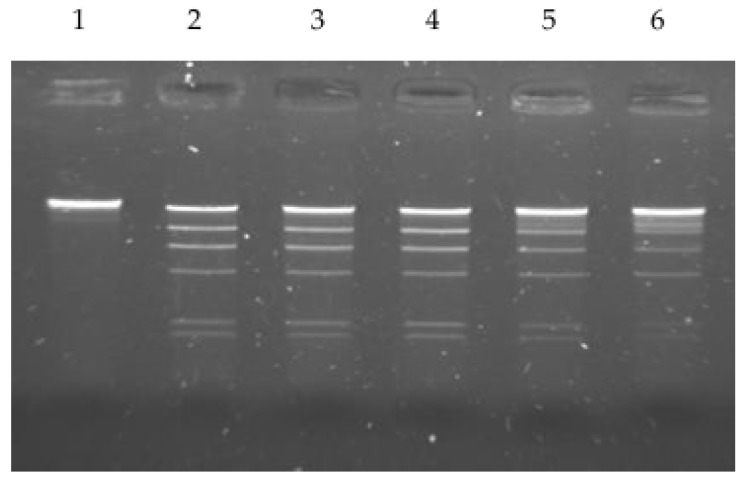
Determination of optimum temperature of the purified PsaI restriction enzyme activity. Digestion of lambda DNA was carried out in NEB2 buffer for 1 h at lane 2 at 30 °C, lane 3 at 37 °C, lane 4 at 42 °C, lane 5 at 52 °C, and lane 6 at 60 °C. Lane 1—undigested lambda DNA.

**Figure 2 ijms-26-06548-f002:**
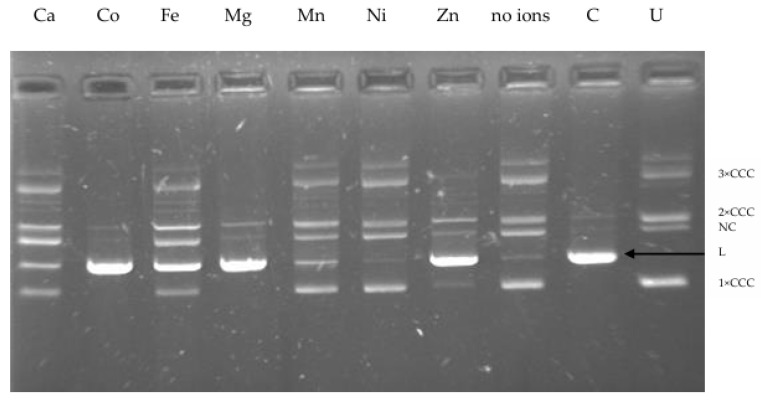
Effect of divalent metal ions on the activity of R.PsaI. Cleavage reactions of pUC18 DNA were carried out in the presence of 10 mM of Ca^2+^, Co^2+^, Fe^2+^, Mg ^2+^,Mn^2+^, Ni^2+^, and Zn^2+^ or without any metal ion. C—control lane containing pUC18 digested with R.PsaI in NEB2 buffer; U—undigested pUC18 DNA, where an arrow indicates the linear form of pUC18 DNA (2686 bp); 3 × CCC—supercoiled plasmid trimer; 2 × CCC—supercoiled plasmid dimer; 1 × CCC—supercoiled plasmid monomer; NC—nicked circular plasmid DNA; L—linear form of pUC18.

**Figure 3 ijms-26-06548-f003:**
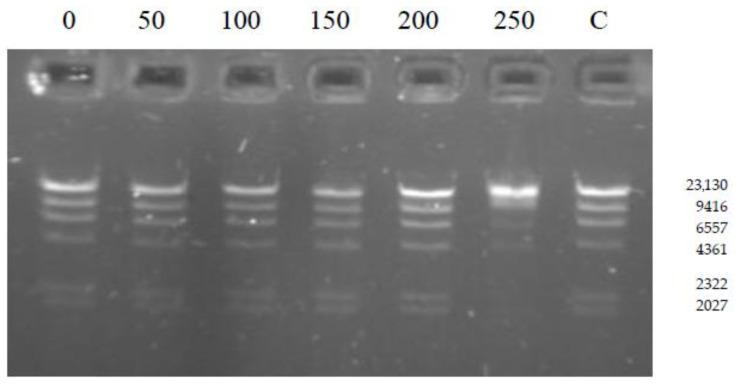
Effect of the NaCl on R.PsaI activity. Cleavage reactions of lambda DNA were carried out in the presence of different NaCl concentrations (0, 50, 100, 150, 200, 250 mM). C—control lane containing lambda DNA incubated with R.PsaI in NEB2 buffer.

**Figure 4 ijms-26-06548-f004:**
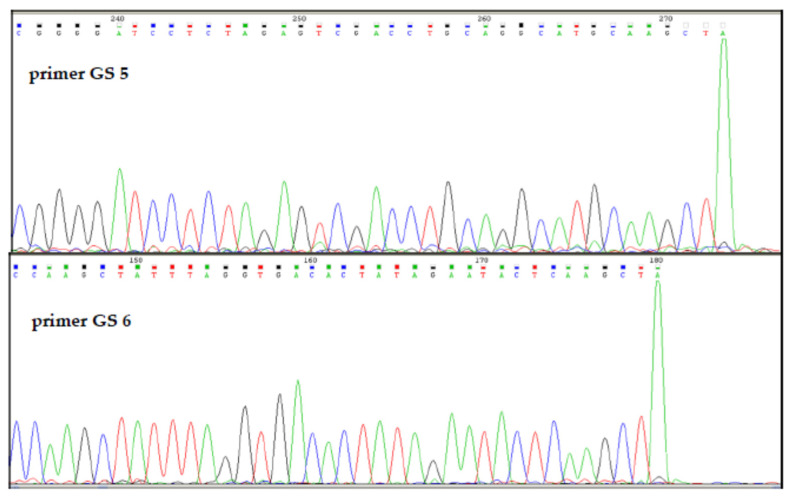
Run-off sequencing to determine the R.PsaI cleavage site in pGEM3Zf(+). The drop in the peak signal indicates where the DNA polymerase runs off of the template at the nicked site. Modified DNA polymerase adds an additional adenine (A) at the end of the extension product.

**Figure 5 ijms-26-06548-f005:**
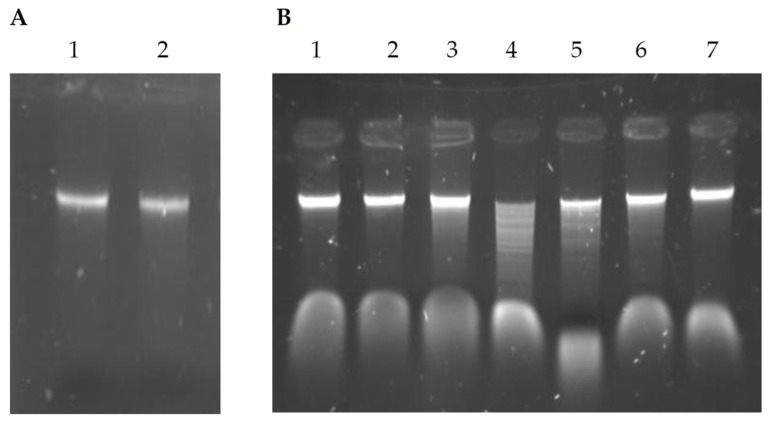
Digestion of the genomic DNA of the *Pseudomonas anguilliseptica* KM9 strain with restriction endonucleases. Panel (**A**): R.PsaI—lane 2; lane 1 shows undigested DNA as a control; panel (**B**): R.HindIII—lane 2, R.EcoRI—lane 3, R.EcoRV—lane 4, R.KpnI—lane 5, R.BamHI—lane 6, and R.EcoVIII—lane 7; lane 1 shows undigested DNA as a control.

**Figure 6 ijms-26-06548-f006:**
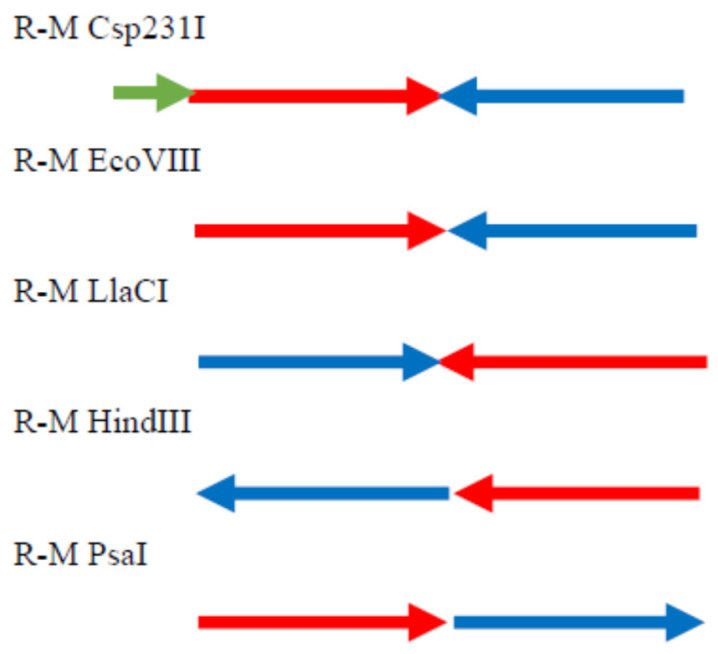
The organization of gene clusters encoding HindIII-like R-M systems. Arrows represent the direction of translation and the relative sizes of open reading frames. The control protein is highlighted in green, restriction endonucleases in red, and methyltransferases in blue.

**Figure 7 ijms-26-06548-f007:**
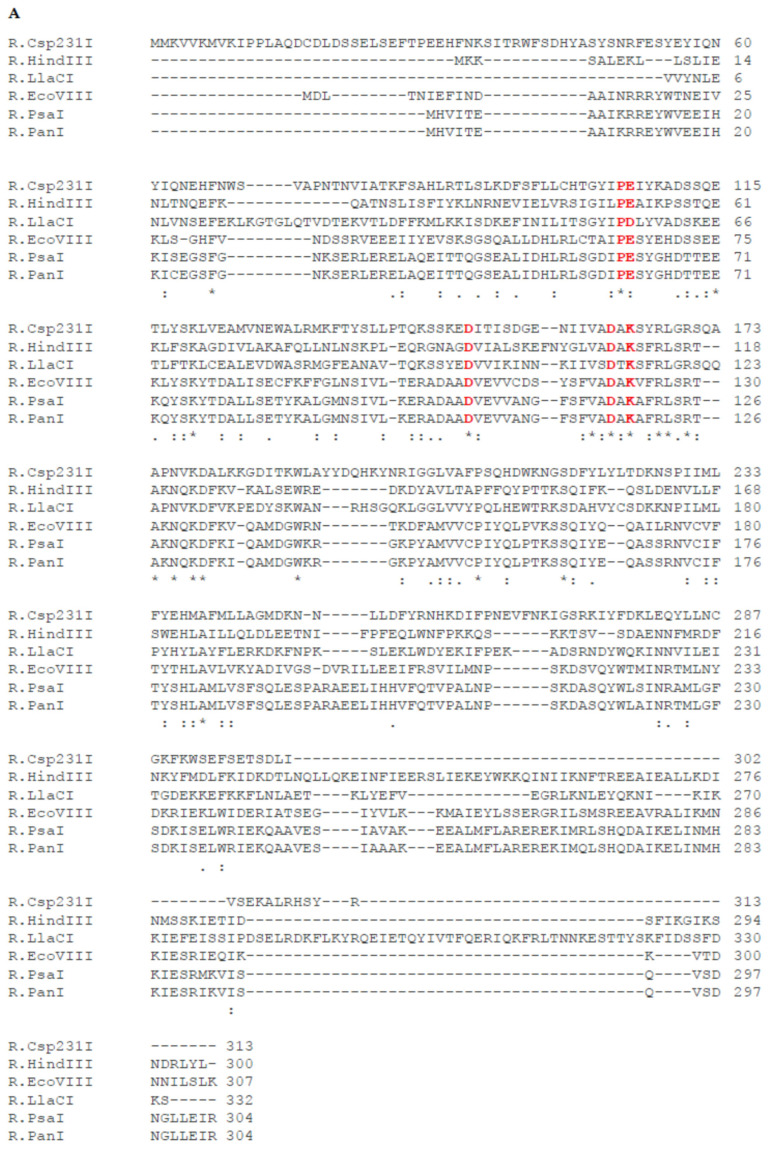
Comparison of the amino acid sequences of restriction endonucleases (**A**) and methyltransferases (**B**) of HindIII-like R-M systems. The amino acids of the putative catalytic magnesium binding motif PD-(D/E)XK are shown in red. The conserved sequence motifs corresponding to the methyltransferase catalytic motif IV (DIPY) and the AdoMet binding motif I (FGG) are marked. Identical amino acids are indicated with an asterisk; two dots represent a highly conservative substitution, and one dot represents a conservative substitution.

**Table 1 ijms-26-06548-t001:** Purification of restriction endonuclease R.PsaI from *Pseudomonas anguilliseptica* KM9.

R.PsaI Purification Step	Total Protein (mg)	Total Activity (U × 10^3^)	Specific Activity (U mg^−1^)	Yield (%)	Purification (Fold)
Cell free extract	652	10,000	15.54 × 10^3^	1	100
Phosphocellulose	48	2920	60.83 × 10^3^	4	29
Hydroxylapatite	13.5	1280	94.81 × 10^3^	6	12.8
CM Sephadex	1.35	450	333.3 × 10^3^	22	4.5

## Data Availability

The data presented in this study are available upon request from the corresponding author.

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
