# Peer review of "Molecular Characterization of a Restriction Endonuclease PsaI from Pseudomonas anguilliseptica KM9 and Sequence Analysis of the PsaI R-M System"

_ijms, 2025, doi:10.3390/ijms26146548_

Round 1
Reviewer 1 Report
Comments and Suggestions for Authors
Dear Authors
The manuscript need improve, I describe some comments and suggestions by section , it is found in the following description:
Abstract
Line 21, please, write the “distinct location”, because do not clear that sites are compared.
Introduction.
Line 78. please add the meaning about “WWTP’S” was not before defined.
Line 81. Eliminated the phrase “Different methods can be used to detect R-M systems” , it is not add anything to the explanation, because the sentence continue to describe the method used in the job.
Line 89 and 104, 146, 191-193. Here describe the (Figure S1) and (Figure S2), but the supplementary materials were not added, check it please.
I suggest reorganize figures the figure S1 and S2, S3 could be add such as principal figures 1 and add panels A), B)... and the figure 1 and 2 could be used the same strategy used the reorganization of panels A) and B) as an only figure 2.
Line 161,172, . was not found the Supplementary materials.
this way is difficult receiving the manuscript, sorry.
Line 160. Here describe that the “plasmid pGEM3Zf(+) was digested with a purified enzyme”, what does enzyme commercial or the R.PsaI obtained?
In the figures 1,2,3, 5 should show the lane ladder marker, please include it.
Lines 285-291. Here described that the region of the R-M complex is at different position genomic…however, it is a supposition because could be performed to experiment to demonstrate it.
In the section 2.6, 2.7, 2.8, described the theories and hypothesis, I suggest to analyze this section and write and describe only the results.
Line 297-300. Please add the sequencing method for to detect blaTem gene or others confirmation identity genes as blaCTX-M, blaTEM, blaSHV and blaAmpC, and intI1, intI2.
The figures gels results should point out the band with size molecular and used arrow.
Line 312-328. Here describe the hypothesis and association to the presence of ESBL to resistance, and blaTEM presence it is suggestion to susceptibility antibiotics, but I suggest used this section for enrich the discussion section, because the manuscript is focus to PsaI enzyme and characterization molecular, but it is not associated to activity of antimicrobial resistant is not demonstrated by presence this enzyme gene.
Discussion
Line 123-126. The star effect, why, was not evaluated?
The discussion section repeat the results again.
This section should be focused to describe the results meaning, for example is common find in this strain Pseudomonas spp the R-N system?, why is it important, is it transfer between strains, what does it ability exceptional give? For example, the % homology sequences analyzed, what is it meaning?
Line 376-380. Here explain the resistance activity to NaCl, however this results and analyzes is not described in the results section, in this case could be explained, why is it resistant to NaCl.
I suggest that sequencing the genome of the isolate for characterized better the ESBL genes and R-M system and 5’- 3’region maybe could describe the organization genomic and even demonstrated the location R-M complex…
Materials and Methods.
section 4.4. here described that exist a integron, it is not describe in the results section and it is not discussed in discussion section, why?
Comments on the Quality of English Language
The English could be improved to more clearly express the research.
Author Response
Thank you very much for your comments. The present form of the manuscript has been modified according to the reviewer’s suggestions. Our answers to the points made by the reviewer (in red) are given below
Reviewer 1
Comments and Suggestions for Authors
Dear Authors
The manuscript need improve, I describe some comments and suggestions by section , it is found in the following description:
Abstract
Line 21, please, write the “distinct location”, because do not clear that sites are compared.
Done according to the reviewer suggestion, line 21
Introduction.
Line 78. please add the meaning about “WWTP’S” was not before defined.
Done according to the reviewer suggestion, line 55
Line 81. Eliminated the phrase “Different methods can be used to detect R-M systems” , it is not add anything to the explanation, because the sentence continue to describe the method used in the job.
Done according to the reviewer suggestion, line87
Line 89 and 104, 146, 191-193. Here describe the (Figure S1) and (Figure S2), but the supplementary materials were not added, check it please.
In the new version of manuscript supplementary materials are included
I suggest reorganize figures the figure S1 and S2, S3 could be add such as principal figures 1 and add panels A), B)... and the figure 1 and 2 could be used the same strategy used the reorganization of panels A) and B) as an only figure 2.
The results presented in figures S1, S2 and S3 are not thematically consistent, so they are presented individually.
Line 161,172, . was not found the Supplementary materials.
this way is difficult receiving the manuscript, sorry.
In the new version of manuscript supplementary materials are included
Line 160. Here describe that the “plasmid pGEM3Zf(+) was digested with a purified enzyme”, what does enzyme commercial or the R.PsaI obtained?
Done according to the reviewer suggestion
In the figures 1,2,3, 5 should show the lane ladder marker, please include it.
Done according to the reviewer suggestion
Lines 285-291. Here described that the region of the R-M complex is at different position genomic…however, it is a supposition because could be performed to experiment to demonstrate it.
We agree with the reviewer that it is speculative to assume that R-M system PsaI is located differently than R-M system PanI. Since this strain has interesting features, to confirm our supposition, we are going to sequence genomic DNA of P. anguilliseptica KM9.
For the purposes of this publication we have added information confirming our assumption in subsection 2.8.
In the section 2.6, 2.7, 2.8, described the theories and hypothesis, I suggest to analyze this section and write and describe only the results.
Following the suggestion of other reviewers, in the new version of the manuscript the results and discussion section have been combined.
Line 297-300. Please add the sequencing method for to detect blaTem gene or others confirmation identity genes as blaCTX-M, blaTEM, blaSHV and blaAmpC, and intI1, intI2.
Following the suggestions of other reviewers, sections 2.9 and 2.10 were removed from the manuscript
The figures gels results should point out the band with size molecular and used arrow.
Done according to the reviewer suggestion
Line 312-328. Here describe the hypothesis and association to the presence of ESBL to resistance, and blaTEM presence it is suggestion to susceptibility antibiotics, but I suggest used this section for enrich the discussion section, because the manuscript is focus to PsaI enzyme and characterization molecular, but it is not associated to activity of antimicrobial resistant is not demonstrated by presence this enzyme gene.
Following the suggestions of other reviewers, we removed this fragment from the manuscript
Discussion
Line 123-126. The star effect, why, was not evaluated?
Under extreme non-standard conditions, such as high ionic strength, restriction endonucleases are known for their ability to cut sequences which are similar but not identical to their defined recognition sequence. This altered specificity called star activity was not the subject of our research
The discussion section repeat the results again.
This section should be focused to describe the results meaning, for example is common find in this strain Pseudomonas spp the R-N system?, why is it important, is it transfer between strains, what does it ability exceptional give? For example, the % homology sequences analyzed, what is it meaning?
In the new version of the manuscript the results and discussion section have been combined. We have added information on the properties of R-M systems and their importance for bacterial cells to the publication, section 2.8
Line 376-380. Here explain the resistance activity to NaCl, however this results and analyzes is not described in the results section, in this case could be explained, why is it resistant to NaCl.
In the new version of the manuscript the results and discussion section have been combined subsection 2.4
I suggest that sequencing the genome of the isolate for characterized better the ESBL genes and R-M system and 5’- 3’region maybe could describe the organization genomic and even demonstrated the location R-M complex…
We thank the reviewer for suggestion regarding Pseudomonas anguilliseptica KM9 genomic DNA sequencing. We intend to do this in the future as the strain has some interesting features and we continue to work on it.
Materials and Methods.
section 4.4. here described that exist a integron, it is not describe in the results section and it is not discussed in discussion section, why?
Following the suggestions of other reviewers, sections 4.4 was removed from the manuscript
Comments on the Quality of English Language
The English could be improved to more clearly express the research.
The manuscript has undergone linguistic proofreading
Reviewer 2 Report
Comments and Suggestions for Authors
This manuscript reported to characterize a restriction endonuclease Psa I from Pseudomonas anguiliseptica KM9, an isoschizomer of Hind III. It was shown that Psa I was purified to approximate homogeneity through 4 chromatographic steps with a yield of 4.5%. The recognition site of Psa I was confirmed as AAGCTT experimentally with the cleavage site between two adenines. Further, the optimal conditions such as temperature, pH, divalent metal ions, etc., were defined. In addition, the genes encoding Psa I and its cognate M.Psa I were amplified by PCR and their amino acid sequences were analysis by multiple alignments, indicating that their sequences were somehow distinct although with the same recognition site. However, this manuscript does not provide novel insights into the type II restriction endonucleases.
Besides, specifical comments also included.
1. The subsection 2.9 and 2.10 are not relative to the manuscript topics, then it is better to be deleted.
2, p.2 L78, The abbreviation “WWTPs” should spelled out at its first occurrence.
3. p.4, L. 136, the authors claimed that “Four metal ions (calcium, iron, manganese and nickel) failed to support cleavage”. But based on Fig. 2, there is a same band even weaker in lane 1 (Ca) and lane 5 (Mn), which is corresponding to the cleavage product (see the attached Figure 2 in PDF). In addition, the DNA bands pattern of pUC18 is somehow confused.
4. p.13-14, L409-410, to specify how to identify the taxonomic classification of P. anguilliseptica KM9 by MALDI-TOF MS, please!
5. The volume unit “L” or “μl” used in the M & M section should be unified.

Author Response
Thank you very much for your comments. The present form of the manuscript has been modified according to the reviewer’s suggestions. Our answers to the points made by the reviewer (in red) are given below
Reviewer 2
Comments and Suggestions for Authors
This manuscript reported to characterize a restriction endonuclease Psa I from Pseudomonas anguiliseptica KM9, an isoschizomer of Hind III. It was shown that Psa I was purified to approximate homogeneity through 4 chromatographic steps with a yield of 4.5%. The recognition site of Psa I was confirmed as AAGCTT experimentally with the cleavage site between two adenines. Further, the optimal conditions such as temperature, pH, divalent metal ions, etc., were defined. In addition, the genes encoding Psa I and its cognate M.Psa I were amplified by PCR and their amino acid sequences were analysis by multiple alignments, indicating that their sequences were somehow distinct although with the same recognition site. However, this manuscript does not provide novel insights into the type II restriction endonucleases.
Besides, specifical comments also included.
- The subsection 2.9 and 2.10 are not relative to the manuscript topics, then it is better to be deleted.
Following the suggestion of reviewer, subsections 2.9 and 2.10 were removed from the manuscript
2, p.2 L78, The abbreviation “WWTPs” should spelled out at its first occurrence.
Done according to the reviewer suggestion, line 55
- p.4, L. 136, the authors claimed that “Four metal ions (calcium, iron, manganese and nickel) failed to support cleavage”. But based on Fig. 2, there is a same band even weaker in lane 1 (Ca) and lane 5 (Mn), which is corresponding to the cleavage product (see the attached Figure 2 in PDF). In addition, the DNA bands pattern of pUC18 is somehow confused.
Plasmid pUC 18 used in this work was isolated and purified using a cesium chloride density gradient centrifugation method. Due to the lack of multimer resolution system, this high copy number vector is typically isolated as a mixture of monomeric and multimeric forms. In the new version of manuscript we marked possible pUC18 forms to facilitate interpretation of experiment.
- p.13-14, L409-410, to specify how to identify the taxonomic classification of P. anguilliseptica KM9 by MALDI-TOF MS, please!
Done according to the reviewer suggestion, subsection 3.2
- The volume unit “L” or “μl” used in the M & M section should be unified.
Done according to the reviewer suggestion
Reviewer 3 Report
Comments and Suggestions for Authors
The manuscript entitled "Molecular Characterization of A Restriction Endonuclease PsaI From Pseudomonas anguilliseptica KM9 and Sequence Analysis of The PsaI R-M System" presents a detailed analysis of the biochemical and genetic properties of the newly isolated restriction enzyme PsaI from the Gram-negative bacterium Pseudomonas anguilliseptica KM9. The authors describe the purification of the enzyme to homogeneity and characterize its activity in the presence of various metal ions, demonstrating high activity with Co²⁺, Mg²⁺, and Zn²⁺, and a significant decrease in activity in the presence of Ca²⁺, Fe²⁺, Mn²⁺, and Ni²⁺. A unique feature of PsaI is that it does not require NaCl for enzymatic activity, which distinguishes it from many other type II endonucleases. DNA sequence analysis confirmed that the enzyme recognizes the 5’-AAGCTT-3’ site, identical to the well-known HindIII enzyme.
The study also characterizes the PsaI restriction-modification system at the genetic level, indicating a high (>98%) sequence similarity with the PanI system from another P. anguilliseptica strain, suggesting the possibility of horizontal gene transfer. An interesting aspect is the identification of the blaTEM gene, encoding β-lactamase, in the genome of the studied strain. Despite the presence of this gene, the strain showed high sensitivity to most of the antibiotics tested, including β-lactams, aminoglycosides, and fluoroquinolones, which the authors attribute to the lack of blaTEM gene expression.
Overall, the manuscript addresses a topic of high relevance and potential scientific impact. While the work is of good quality and demonstrates application potential, it would benefit from major revisions, particularly in the absence of supplementary material and in terms of editorial clarity, to ensure its findings are fully accessible and impactful for the research community. With these improvements, the manuscript has the potential to make a significant contribution to the research.
Details below:
Comment 1#
The submission did not include any supplementary materials. The authors refer to tables and figures that the reviewer does not see (line 89, 104, 146, 161, 172, 193, 204, 223, 436)
Comment 2#
Line 10: The correct form is "Gram-negative"
Comment 3#
Line 78: Explain the abbreviation you are using in the text for the first time (WWTPs)
Comment 4#
No marker on any of the electrophoretic gels.
Comment 5#
Figure 5: Place a caption directly under the figure.
Comment 6#
Figure 8: No marked values on the DNA marker
Comment 7#
It seems to me that a combination of results and discussion would have made more sense in this work. In the results, the authors cite a lot of work based on available publications anyway, which is already somewhat of a discussion.
Comment 8#
Revise references according to the journal's guidelines.
Author Response
Thank you very much for your comments. The present form of the manuscript has been modified according to the reviewer’s suggestions. Our answers to the points made by the reviewer (in red) are given below
Reviewer 3
Comments and Suggestions for Authors
The manuscript entitled "Molecular Characterization of A Restriction Endonuclease PsaI From Pseudomonas anguilliseptica KM9 and Sequence Analysis of The PsaI R-M System" presents a detailed analysis of the biochemical and genetic properties of the newly isolated restriction enzyme PsaI from the Gram-negative bacterium Pseudomonas anguilliseptica KM9. The authors describe the purification of the enzyme to homogeneity and characterize its activity in the presence of various metal ions, demonstrating high activity with Co²⁺, Mg²⁺, and Zn²⁺, and a significant decrease in activity in the presence of Ca²⁺, Fe²⁺, Mn²⁺, and Ni²⁺. A unique feature of PsaI is that it does not require NaCl for enzymatic activity, which distinguishes it from many other type II endonucleases. DNA sequence analysis confirmed that the enzyme recognizes the 5’-AAGCTT-3’ site, identical to the well-known HindIII enzyme.
The study also characterizes the PsaI restriction-modification system at the genetic level, indicating a high (>98%) sequence similarity with the PanI system from another P. anguilliseptica strain, suggesting the possibility of horizontal gene transfer. An interesting aspect is the identification of the blaTEM gene, encoding β-lactamase, in the genome of the studied strain. Despite the presence of this gene, the strain showed high sensitivity to most of the antibiotics tested, including β-lactams, aminoglycosides, and fluoroquinolones, which the authors attribute to the lack of blaTEM gene expression.
Overall, the manuscript addresses a topic of high relevance and potential scientific impact. While the work is of good quality and demonstrates application potential, it would benefit from major revisions, particularly in the absence of supplementary material and in terms of editorial clarity, to ensure its findings are fully accessible and impactful for the research community. With these improvements, the manuscript has the potential to make a significant contribution to the research.
Details below:
Comment 1#
The submission did not include any supplementary materials. The authors refer to tables and figures that the reviewer does not see (line 89, 104, 146, 161, 172, 193, 204, 223, 436)
In the new version of manuscript supplementary materials are included
Comment 2#
Line 10: The correct form is "Gram-negative"
Done according to the reviewer suggestion, line 10
Comment 3#
Line 78: Explain the abbreviation you are using in the text for the first time (WWTPs)
Done according to the reviewers suggestion, line 55
Comment 4#
No marker on any of the electrophoretic gels.
We have marked the size of DNA/protein markers
Comment 5#
Figure 5: Place a caption directly under the figure.
Done according to the reviewer suggestion
Comment 6#
Figure 8: No marked values on the DNA marker
We have marked the size of DNA marker
Comment 7#
It seems to me that a combination of results and discussion would have made more sense in this work. In the results, the authors cite a lot of work based on available publications anyway, which is already somewhat of a discussion.
We have combined results and discussion sections according to the reviewer suggestions.
Comment 8#
Revise references according to the journal's guidelines.
We have revised references according to the journal’s guidelines
Reviewer 4 Report
Comments and Suggestions for Authors
This study by Furmanek-Blaszk et al provides a genetic and biochemical characterisation of PsaI, a type II restriction enzyme. The purified the enzyme in functional form and elucidated its recognition sequence, dependence of its activity on the divalent cations Co2+, Mg2+ and Zn2+ but inhibited by Ca2+, Fe2+, Mn2+ and Ni2+ and its independence on the monovalent Na+.
Line 53 - Could the authors exemplify those specificities that have not been found?
Line 102: the Coomassie gel not shown?
Figure 1= It will be informative to have reaction conditions such as incubation time, buffer used, in this figure legend.
Figure 2: could the authors explain the reason for multiple bands in the lanes for undigested plasmid pUC18? In the present form, it is difficult to make a good sense of information from the agarose gel, and the figure legend/text does not explain what band correspond to cut or uncut DNA. What is the size of the uncut and the expected fragment(s)? Are there supercoiled DNA band on the gel? There is no DNA ladder/molecular weight marker as standard to help the reader.
I suggest that the author use densitometry or similar approaches to quantify the intensity of bands such that the impact of the studied variables comes out crystal-clear.
Line 145-153 – why does activity failed in high ionic strength? Does the sample viscosity increased or electrostatic screening preventing formation of enzyme-DNA intermediate complex?
Line 174 – what results in smearing from using an enzyme with “different specificities”?
Author Response
Thank you very much for your comments. The present form of the manuscript has been modified according to the reviewer’s suggestions. Our answers to the points made by the reviewer (in red) are given below
Reviewer 4
Comments and Suggestions for Authors
This study by Furmanek-Blaszk et al provides a genetic and biochemical characterisation of PsaI, a type II restriction enzyme. The purified the enzyme in functional form and elucidated its recognition sequence, dependence of its activity on the divalent cations Co2+, Mg2+ and Zn2+ but inhibited by Ca2+, Fe2+, Mn2+ and Ni2+ and its independence on the monovalent Na+.
Line 53 - Could the authors exemplify those specificities that have not been found?
There is a need for constant seeking of new restriction enzymes that recognize specific DNA sequences and which expands the tools for genetic engineering, line 51
Line 102: the Coomassie gel not shown?
Coomassie gel is in supplementary materials
Figure 1= It will be informative to have reaction conditions such as incubation time, buffer used, in this figure legend.
Done according to the reviewer suggestion
Figure 2: could the authors explain the reason for multiple bands in the lanes for undigested plasmid pUC18? In the present form, it is difficult to make a good sense of information from the agarose gel, and the figure legend/text does not explain what band correspond to cut or uncut DNA. What is the size of the uncut and the expected fragment(s)? Are there supercoiled DNA band on the gel? There is no DNA ladder/molecular weight marker as standard to help the reader.
I suggest that the author use densitometry or similar approaches to quantify the intensity of bands such that the impact of the studied variables comes out crystal-clear.
Plasmid pUC 18 used in this work was isolated and purified using a cesium chloride density gradient centrifugation method. Due to the lack of multimer resolution system, this high copy number vector is typically isolated as a mixture of monomeric and multimeric forms. In the new version of manuscript we marked possible pUC18 forms to facilitate interpretation of experiment.
Line 145-153 – why does activity failed in high ionic strength? Does the sample viscosity increased or electrostatic screening preventing formation of enzyme-DNA intermediate complex?
Many restriction enzymes prefer salt concentration between 50-150mM, increased salt concentration in buffer can inhibit enzyme activity. High concentrations of salt (like NaCl) affect the enzyme’s interaction with DNA, line175-188
Line 174 – what results in smearing from using an enzyme with “different specificities”?
In this experiment we showed that the functional R-M system present in the Pseudomonas anguilliseptica KM9 strain protects genomic DNA against his endogenous restriction enzyme while enzymes with different specificities degrade DNA. This is one of the methods of detecting functional R-M systems, line 200-215
Round 2
Reviewer 1 Report
Comments and Suggestions for Authors
Dear authors
The new version is better and complete, it attended the comments and suggestion or you give the justification why not include the suggestions, which I consider valid.
However, The supplemental material I did not have access for its reviewing. I consider thta the editor could be check it and if it did not has any mistake, them, I consider that the manuscript could be suggest for publish.
Author Response
Comments and Suggestions for Authors
Dear authors
The new version is better and complete, it attended the comments and suggestion or you give the justification why not include the suggestions, which I consider valid.
However, The supplemental material I did not have access for its reviewing. I consider thta the editor could be check it and if it did not has any mistake, them, I consider that the manuscript could be suggest for publish.
We appreciate the time and effort you invested in reviewing our paper.
We have carefully considered all of your comments and incorporated the necessary changes.
With regard to the supplementary file, we are sorry that you have problems with this. However, everything from our end appears to be in good order. Please could you ask the editor for help.
Reviewer 2 Report
Comments and Suggestions for Authors
This version is improved greatly with the review comments to be fully considered. I suggest the manuscript could be accepted except a minor issue in Fig. 5, in which it is better to add an extra lane with the genomic DNA digested by Psa I.
Author Response
Comments and Suggestions for Authors
This version is improved greatly with the review comments to be fully considered. I suggest the manuscript could be accepted except a minor issue in Fig. 5, in which it is better to add an extra lane with the genomic DNA digested by Psa I.
We agree with your suggestion regarding Figure 5. We have made the changes you recommended.
Reviewer 3 Report
Comments and Suggestions for Authors
The authors have complied with the suggested comments. The work looks much better, but in future, please mark all changes made to the text in red. Good luck in your further scientific work.
Author Response
Comments and Suggestions for Authors
The authors have complied with the suggested comments. The work looks much better, but in future, please mark all changes made to the text in red. Good luck in your further scientific work.
Thank you for your valuable feedback, which has helped us improve the manuscript.
Reviewer 4 Report
Comments and Suggestions for Authors
The manuscript has imporved and the figure legends are more detailed making the results easier to follow.
Author Response
Comments and Suggestions for Authors
The manuscript has imporved and the figure legends are more detailed making the results easier to follow.
Thank you for your valuable feedback, which has helped us improve the manuscript.